# Anxiety among Adolescents and Young Adults during COVID-19 Pandemic: A Multi-Country Survey

**DOI:** 10.3390/ijerph191710538

**Published:** 2022-08-24

**Authors:** Heba Jafar Sabbagh, Wafaa Abdelaziz, Waleed Alghamdi, Maryam Quritum, Nada AbuBakr AlKhateeb, Joud Abourdan, Nafeesa Qureshi, Shabnum Qureshi, Ahmed H. N. Hamoud, Nada Mahmoud, Ruba Odeh, Nuraldeen Maher Al-Khanati, Rawiah Jaber, Abdulrahman Loaie Balkhoyor, Mohammed Shabi, Morenike Oluwatoyin Folayan, Omolola Alade, Noha Gomaa, Raqiya Alnahdi, Nawal A. Mahmoud, Hanane El Wazziki, Manal Alnaas, Bahia Samodien, Rawa A. Mahmoud, Nour Abu Assab, Sherin Saad, Sondos G. Alhachim, Maha El Tantawi

**Affiliations:** 1Department of Pediatric Dentistry, Faculty of Dentistry, King Abdulaziz University, Jeddah 21589, Saudi Arabia; 2Department of Pediatric Dentistry and Dental Public Health, Faculty of Dentistry, Alexandria University, Alexandria 21561, Egypt; 3Division of Psychiatry, Faculty of Medicine, King Abdulaziz University, Jeddah 21589, Saudi Arabia; 4Faculty of Medicine, King Abdulaziz University, Jeddah 21589, Saudi Arabia; 5Medical Faculty, Istanbul Medipol University, Istanbul 34230, Turkey; 6City Quay Dental Practice and Implant Centre, Dundee DD1 3JA, UK; 7Department of Education, University of Kashmir, Srinagar 190006, India; 8Faculty of Dentistry, Alexandria University, Alexandria 21561, Egypt; 9Faculty of Dentistry, National Ribat University, Khartoum 1111, Sudan; 10College of Dentistry, Ajman University, Ajman P.O. Box 346, United Arab Emirates; 11Department of Oral and Maxillofacial Surgery, Faculty of Dentistry, Syrian Private University, Damascus 368, Syria; 12General Courses, King Abdulaziz University, Jeddah 21589, Saudi Arabia; 13Faculty of Engineering, King Abdulaziz University, Jeddah 21589, Saudi Arabia; 14University of Jeddah, Jeddah 23218, Saudi Arabia; 15Department of Child Dental Health, Obafemi Awolowo University, Ile-Ife, Nigeria; 16Department of Preventive and Community Dentistry, Faculty of Dentistry, Obafemi Awolowo University, Ile-Ife, Nigeria; 17Schulich School of Medicine and Dentistry, University of Western Ontario, London, ON N6A 3K7, Canada; 18Department of Dental Surgery, Oman Dental College, Muscat 116, Oman; 19Institute of Creative Art and Design (ICAD), Kuala Lumpur Campus, UCSI University, Kuala Lumpur 56000, Malaysia; 20Department of Cereal Plant Pathology, National Institute of Agricultural Research, Rabat 10090, Morocco; 21Division of Imaging Science and Technology, School of Medicine, University of Dundee, Dundee DD1 4HN, UK; 22Western Cape Education Department, Cape Town 8001, South Africa; 23Musculoskeletal Center, International Medical Center, Jeddah 21451, Saudi Arabia; 24Schools of Awqaf, Directorate of Education, Jerusalem, Israel; 25Department of Pharmacology, Institute of Neuroscience and Physiology, University of Gothenburg, Box 431, 40530 Gothenburg, Sweden; 26Health Education Services, Ingham County, Lansing, MI 48933, USA

**Keywords:** adolescent and young adults, anxiety, country income level, socio-demographic, COVID-19

## Abstract

(1) Background: Adolescents-and-young-adults (AYA) are prone to anxiety. This study assessed AYA’s level of anxiety during the COVID-19 pandemic; and determined if anxiety levels were associated with country-income and region, socio-demographic profile and medical history of individuals. (2) Methods: A survey collected data from participants in 25 countries. Dependent-variables included general-anxiety level, and independent-variables included medical problems, COVID-19 infection, age, sex, education, and country-income-level and region. A multilevel-multinomial-logistic regression analysis was conducted to determine the association between dependent, and independent-variables. (3) Results: Of the 6989 respondents, 2964 (42.4%) had normal-anxiety, and 2621 (37.5%), 900 (12.9%) and 504 (7.2%) had mild, moderate and severe-anxiety, respectively. Participants from the African region (AFR) had lower odds of mild, moderate and severe than normal-anxiety compared to those from the Eastern-Mediterranean-region (EMR). Also, participants from lower-middle-income-countries (LMICs) had higher odds of mild and moderate than normal-anxiety compared to those from low-income-countries (LICs). Females, older-adolescents, with medical-problems, suspected-but-not-tested-for-COVID-19, and those with friends/family-infected with COVID-19 had significantly greater odds of different anxiety-levels. (4) Conclusions: One-in-five AYA had moderate to severe-anxiety during the COVID-19-pandemic. There were differences in anxiety-levels among AYAs by region and income-level, emphasizing the need for targeted public health interventions based on nationally-identified priorities.

## 1. Introduction

Generalized anxiety disorders refer to uncontrollable fear, worry excessive apprehension, and hyperarousal [1]. It is the most common mental disorder among children and adolescents [2] with estimated prevalence ranging between 0.9% and 28.3%, 45.82 million incident cases and 301.39 million prevalent cases [3,4]. Global research on adolescents, including 82 countries with wide geographic variety and cultural background, reported anxiety prevalence ranging from 7% to 12% [5]. Furthermore, a systematic review searched data from 1990 to 2019, and reported a 50% increase in the annual percentage anxiety change. They also reported a vast differences and trend in anxiety in relation to gender, age and nation [4]. Other studies reported variations in anxiety prevalence according to geographic area, cultural differences especially in family and peer relationships [5] which differ worldwide and by country development [2,3,6]. These differences and changes in anxiety obligate evaluation of recent changes, especially after the COVID-19 pandemic. In addition, evaluating the financial, regional and socio-demographic background is important to develop policies tailored for each nation.

Adolescence is a critical period as it considered a transformational where children are growing to adulthood and their immaturity is developing to maturity making them vulnerable to external and environmental pressures [7,8]. Anxiety is associated with the period of adolescence emphasizing the public health importance of this problem [9,10]. Adolescents and young adults (AYA) quality of life, general and oral health are affected by their level of anxiety [11,12]. Anxiety negatively impacts AYAs’ educational achievement, self-esteem, and may result in lifelong impairment [7]. Also, anxiety increases the risk of oral diseases and decreases the utilization of oral health care among AYA [13,14]. 

COVID-19 pandemic was found to be associated with the risk of anxiety [15]. It has resulted in social isolation, restricted peer interactions, and reduced external support from teachers and coaches [16]. Factors associated with high anxiety during the pandemic include fatigue, loneliness, stress, pandemic-related concerns, and changes in health behaviors [17].

A systematic review of 21 studies from China and two from Turkey reported 26% prevalence of anxiety [18]. Furthermore, a meta-analysis of studies from China, Jordan, the United states, Brazil, Greece, Canada, Italy, Spain, and Germany showed that 20.5% of children and adolescents have experienced anxiety during COVID-19; a figure that was double that which existed before the pandemic [19]. However, these systematic reviews included studies from mostly western countries which limits their generalizability.

The physical and psychological condition of adolescents and young adults may have received less attention than other age groups especially during the early stages of the pandemic. However, as the pandemic progressed, concerns were raised about their vulnerability including the risk of mental problems [20]. In addition, previous global research that evaluated adolescent’s anxiety reported similar levels of anxiety among countries with similar cultural and development backgrounds [5]. This was attributed to shared patterns of peer and parental relationships. However, little is known about whether country level factors affect the risk of anxiety among AYA especially during the COVID-19 pandemic. Nevertheless, evaluating mental health in young age groups, such as AYA in different populations allows health care providers and community health services to work on disease prevention and formulating effective prevention program.

Therefore, it is important to fill the knowledge gap by identifying which country versus individual level factors explain differences and similarities in anxiety levels. This study assessed AYA’s level of anxiety during the COVID-19 pandemic; and determined if the anxiety levels were associated with the country-income level, and the socio-demographic profile and medical history of individuals. The null hypothesis of the study was that the severity of anxiety in AYA was not associated with country context, socio-demographic background or medical history including COVID-19 status.

## 2. Materials and Methods

A multi-country survey collected data from adolescents aged 11 to 23 years between August-2020 and January 2021 during the COVID-19 pandemic. Approval for the study was obtained from the Research Ethics Committee of the Faculty of Dentistry, King Abdulaziz University, Jeddah, Saudi Arabia (REC No. 90-08-20; approved on 31 August 2020) and from Health Research Ethic Committee, Institute of Public Health, Obafemi Awolowo University, Ile-Ife, Nigeria (IPH/OAU/12/1604; approved on 8 February 2021). The study was conducted in accordance with the Helsinki declaration [21]. 

Participants were invited to respond to an online questionnaire after the purpose of the study was explained to them and they were assured of the confidentiality of their responses and that they were free to withdraw from the survey at any time. At the beginning of the online survey, participants were required to select one option indicating their age: 11–14, 15–17 and 18–23 [22]. Participants who selected the first two age groups were directed to a parental consent form and after it was filled, the parent was asked to invite the child to respond to the survey without supervision and to assure him/ her of the anonymity and confidentiality of the responses. Respondents older than 18 years proceeded to the survey after consenting by ticking a checkbox. Individuals who did not consent to participate were thanked for their interest and exited from the survey. 

The questionnaire consisted of three sections (see Appendix B). The first section asked respondents about their age at the last birthday, sex at birth and educational level (elementary school, middle school, high school, university) and the mother and fathers’ educational levels (no formal education, finished elementary school, finished middle school, finished high school, finished university or more). 

The second section asked about the participants’ medical history. Participants were required to answer ‘yes’ or ‘no’ about whether they had any medical problem, had confirmed COVID-19 infection, suspected to be infected with COVID-19, and whether a family member or a friend was infected with COVID-19. The COVID-19 questions were adopted from those used for the Mental Health and Wellness Study [23].

The third section assessed the general anxiety level using the Generalized Anxiety Disorder 7-item scale (GAD-7). The GAD-7 describes the frequency of seven items assessing general anxiety on a 4-point scale ranging from 0: not at all, 1: several days, 2: more than half the days to 3: nearly every day. The total score is obtained by adding the points of the seven items and ranges from 0 to 21. Cut off points of 5, 10 and 15 indicate mild, moderate and severe anxiety respectively [24]. The GAD-7 had been validated for use in many of the countries included in this survey with Cronbach alpha ranging from 0.84 to 0.946 [25,26]. For this study, the Cronbach alpha was 0.897.

The survey data collection tool was originally developed and validated in English and Arabic [24,27,28]. In this study, the tool was translated to French, Turkish and Malay by native speaking collaborators. The tool was then tested for validity in the five languages (English, Arabic, French, Turkish and Malay) by calculating the Content validity index (CVI) [29]. The CVI for the Arabic and English versions was 0.87 based on the evaluation by nine dentists; 0.97 for the Turkish version based on the evaluation conducted by seven dentists; 0.80 for the Malay version based on the evaluation conducted by five dentists; and 0.877 for the French version based on the evaluation conducted by five dentists. Each version was further pilot tested with 10 participants to ensure clarity and appropriate use of terms. After testing, the questionnaire was uploaded to the online platform SurveyMonkey^®^, the settings were modified to ensure that the responses were anonymous, and to prevent the submission of multiple responses from the same electronic device. 

The core team from King Abdulaziz University, Saudi Arabia and Alexandria University, Egypt invited collaborators in their network to collect data in different countries. Interested collaborators received the study proposal, ethical approval, and instructions about the target group, sampling strategy, sample size and timeline for the study. A customized SurveyMonkey^®^ data collection link and a recruitment tracking link was made for each collaborator. Snowball sampling was used where each collaborator asked people in their own network to further disseminate the survey link.

### Statistical Analysis

Frequencies and percentages were calculated. Countries were categorized according to the World Health Organization list of regions into: the African Region (AFR), Region of the Americas (AMR), Southeast Region (SEAR), European region (EUR), and Eastern Mediterranean Region (EMR), and Western Pacific Region (WPR). Countries were also classified according to the economic level following the World Bank classification based on the Atlas method (World Bank Country and Lending Groups 2021) into low-income countries (LICs) with gross national income (GNI) ≤ USD 1035 in 2019, lower-middle income countries (LMICs) with GNI between USD 1036 and 4045, upper middle-income countries (UMICs) with GNI between USD 404 and 12,535 and high-income countries (HICs) with GNI ≥ USD 12,536.

The associations between general anxiety levels and country-level factors (region and income), personal factors (age at last birthday, sex at birth and mothers’ and fathers’ educational level), medical history and COVID-19 status were assessed using chi squared test. To accommodate the clustering of participants within countries, multivariable multi-level multinomial logistic regression model was used to assess the association between the dependent variable (mild, moderate and severe versus normal anxiety) and the independent variables (personal, medical and COVID-19-status, WHO region and country income level) which were mutually adjusted for each other. Because of collinearity between participant’s age and educational level, participant’s educational level was removed from the model. Level 1 factors were individual factors and level 2 factors were country level factors. Both t ypes were introduced as fixed effect factors. Countries were introduced as random effect factors with intercepts. Robust estimation was used to handle violations to model assumptions. We used least significance difference ti adjust for multiple testing. Adjusted odds ratios (AORs) and 95% confidence intervals (CIs) were calculated. IBM SPSS for Windows version 22.0 (IBM Corp., Armonk, NY, USA) was used for statistical analysis. Significance was set at 5%.

## 3. Results

Complete responses were available for 6989 AYA from 25 countries (Appendix A). Table 1 shows that most participants were from the EMR region (64.8%), from HICs (38.3%), females (57.3%), young adults aged 18–23 years (74.4%), and with university education (56.8%). The greatest percentage had mothers (42.6%) and fathers (53.3%) with university or higher education. Also, 900 (12.9%) reported having a medical condition, 529 (7.6%) had a history of COVID-19, 1268 (18.1) suspected they had COVID-19 but were not tested; and 3044 (43.6) reported they had a family member or a friend infected or suspected of COVID-19. Also, 902 (12.9%) had moderate anxiety and 503 (7.2%) had severe anxiety. The highest percentage of AYA participants with severe general anxiety disorders were resident in Jordan (20%) and the United Arab Emirates (15.3%) See Appendix A.

There were significantly more participants with severe GAD from WPR countries than from AFR, AMR and SEAR countries (10.2%, 2.3%, 3.9% and 4.9%, *p* < 0.0001). In addition, significantly more participants with severe GAD were from UMICs than LMICs (9.4% and 5.7%, *p* = 0.004); females than males (8.5% and 5.5%, *p* < 0.0001); of older than younger ages (8.4% (18–23years), 4.6% (15 to 17 years) and 2.3% (11 to 14 years), *p* < 0.0001); and with university education than high school, middle and elementary school education (8.7%, 6.4%, 2.7% and 0.8%, *p* < 0.0001). Severe GAD was also significantly associated with higher mothers’ (*p* < 0.0001) and fathers’ (*p* = 0.001) educational level. Also, significantly more participants with medical conditions (12.7% and 6.4%), a history of COVID-19 infection (9.1% and 7.1%), suspected but not tested COVID-19 (11.6% and 6.2%) and with family or friend infected or suspected of COVID-19 (8.5% and 6.2%, *p* < 0.0001) had severe GAD. 

Table 2 shows the result of the multilevel multinomial logistic regression model for factors associated with GAD severity levels. Compared to participants from the EMR, participants from the AFR had significantly lower odds of having mild (AOR = 0.34), moderate (AOR = 0.44) and severe (AOR = 0.46) GAD than normal anxiety, while the odds of having mild, moderate and severe GAD than normal anxiety were not significantly different for participants from EURO compared to those from EMR countries. 

Participants from LMICs had significantly higher odds for mild (AOR = 1.47) and moderate (AOR = 1.49) GAD than normal anxiety compared to participants from LICs. The odds of having different levels of GAD than normal anxiety did not significantly differ for participants from UMICs and HICs compared to LICs.

Females had significantly higher odds of mild (AOR = 1.35), moderate (AOR = 1.39) and severe (AOR = 1.33) GAD than normal anxiety compared to males. 

In addition, Participants who were 11 to14 or 15 to17 years old had significantly lower odds of mild (AOR = 0.68 and AOR = 0.81) and severe (AOR = 0.65 and AOR = 0.76) GAD compared to normal when compared to 18 to23 years old. 

Having medical problems was significantly associated with mild (AOR = 1.53), moderate (AOR = 1.65) and severe (AOR = 1.75) anxiety compared to normal. Higher odds of mild (AOR = 1.35), moderate (AOR = 1.25) and severe (AOR = 1.48) GAD than normal anxiety were significantly associated with suspected COVID-19 infection that was not confirmed by testing. Having friends or family with confirmed or suspected COVID-19 infection was associated with significantly higher odds of mild (AOR = 1.22) and moderate (AOR = 1.24) than normal anxiety.

## 4. Discussion

The study showed that AYA in EMR countries had greater odds of mild than normal anxiety compared to those in countries in other regions except EUR and greater odds of moderate and severe than normal anxiety compared to participants in AFR. Also, AYA in LMICs had significantly higher odds of mild and moderate GAD than those in LICs. Females and 18 to 23 years old group had significantly higher levels of GAD than males and adolescents. Participants with medical problems, those who suspected themselves to have COVID-19 but had not taken a test, and those who had a friend or a family member who had COVID-19 or was suspected to have COVID-19 had significantly higher odds of more severe forms of GAD. The study’s null hypothesis was therefore not supported. 

This study has several strengths. First, participants were included from a wide geographical area ensuring that the impact of the pandemic on the level of anxiety among AYA is studied across different cultures and sociodemographic backgrounds [30]. In addition, the study assessed the impact of country level factors on anxiety levels. Thus, our study filled a knowledge gap by offering a wider perspective that considered macro and micro levels factors associated with generalized anxiety disorder among AYA. The literature on the global prevalence and burden of anxiety during COVID-19 pandemic showed a greater change in anxiety levels in younger ages compared to adults, and a stronger effect of the pandemic on adolescents’ mental health [31,32,33] reinforcing the importance of this study. Differences among countries in healthcare system structure, healthcare seeking behaviors stigmatization of mental problems and others are complex factors that need to be comprehensively studied following the suggested associations observed in the present study.

However, there were some limitations. This was a non-probability sampling process, and an internet based electronic survey due to the pandemic restrictions which may have skewed the respondents to those with university education. Also, anxiety was self-reported and this may lead to the over-estimation of the prevalence of anxiety. Furthermore, there is the chance that adolescents might have consented instead of their parents or claimed to be older to respond on their own, and older respondents may have claimed to be younger to answer the survey. However, at the time this survey was conducted, the enthusiasm to participate in online surveys was already reducing thereby reducing the risk for falsification of age to participate. Finally, this study used a cross-sectional design and, thus, causality cannot be established. Despite these limitations, the study provided important information.

First, AFR had lower odds of severe levels of GAD than EMR. A prior systematic review of the effect of COVID-19 pandemic on global anxiety and depressive disorders showed that the EMR had the highest frequency of general anxiety compared to other regions [31]. Our study confirmed the finding of the review and also indicated that the finding also applied to AYA. This is supported by a study that evaluated generalized anxiety among 6-countries which reported a significant cross-cultural and country-wise differences; highlighting the importance of research that assesses differences and similarities among countries [34].

Also, this study showed that LMICs had higher odds of mild and moderate than normal general anxiety compared to LICs. The proportion of people reported to have COVID-19 was low in LICs compared to other countries and thus, there may have been less anxiety or perception of risk of contracting the infection [35]. The level of anxiety may, however, have been higher in LMICs where the prevalence was higher but the facilities for care were less available than in UMICs or HICs. Moreover, COVID-19 lockdown may have had more severely affected the economics of LMICs exposing them to mental distress [36]. Furthermore, the COVID-19 pandemic could have affected the care of patients with mental illness among LMICs either by creating a barrier to reach health facilities for fear of COVID-19 infection or due to redirecting mental health care facilities for COVID-19 care leading to greater severity of anxiety [37,38].

Second, females showed higher levels of GAD than males in agreement with prior studies [19,39]. This sex predilection for general anxiety might be attributed to biological susceptibilities, lower self-esteem or greater exposure to trauma [40]. In addition, females may be more likely to develop internalizing symptoms such as anxiety following exposure to stressors [41]. 

Third, older AYA were more likely to have mild and severe anxiety than younger persons. This finding disagrees with a systematic review during the pandemic that showed no association between age and COVID-19 related anxiety [19]. It is possible that the young adults included in the present study were more affected than adolescents by the pandemic and its consequences such as closure of universities and social isolation. Also, young adults may be more aware of the risks of the pandemic and its impact on their family’s livelihood which may have contributed to higher levels of anxiety. 

Fourth, a history of medical problems or suspected COVID-19 infection were associated with higher odds of more severe forms of generalized anxiety which agrees with another study conducted during the pandemic [42]. Medical problems make people more vulnerable to serious complications of COVID-19 and this may explain the higher odds of greater anxiety. Also, COVID-19 infection among friends or family members is associated with more worries and distress given the uncertain nature of this illness and its serious consequences [43,44]. Furthermore, Gale et al. (2016) concluded in their systematic review that anxiety among AYA is highly related to their family and friends’ health [45]. The present study also agrees with a 2022 systematic review that assessed the mental disorders of communities following epidemics, including the COVID-19 pandemic. The authors reported poorer mental health for those with higher risk of infection, those with medical conditions, and those showing COVID-19-like symptoms [46].

## 5. Conclusions

The results of this multicounty study suggest that the COVID-19 pandemic was associated with greater probability of anxiety for AYA who were older, females, with medical problems, with positive COVID-19 infection, and that it differed among populations from different regions and income levels. This provides some evidence that can guide the development of care programs for AYA who have mental health challenges during the COVID-19 pandemic. This is especially important considering the need to prioritize the limited resources available for during and post pandemic care. Further studies are needed to explore the age disparity we reported. It is also important to periodically assess the change in the prevalence of anxiety and other mental disorders as the pandemic unfolds so that accurate estimates can be obtained to support the decision making. 

## Figures and Tables

**Table 1 ijerph-19-10538-t001:** Anxiety/GAD level, Country level factors, sociodemographic profile, medical history and COVID-19 status among 11 to 23-year-old individuals from different countries during the second wave of COVID-19 (*n* = 6989).

Factors		Anxiety/GAD Level					
		Normal	Mild	Moderate	Severe	Total	*p* Value
		2964 (42.4%)	2621 (37.5%)	900 (12.9%)	504 (7.2%)		
Country region	EMR	1725 (38.1)	1829 (40.4)	623 (13.8)	349 (7.7)	4526 (64.8)	<0.0001
	AMR	192 (75.0)	49 (19.1)	5 (2.0)	10 (3.9)	256 (3.7)	
	SEAR	208 (42.4)	191 (38.9)	68 (13.8)	24 (4.9)	491 (7.0)	
	WPR	164 (46.5)	102 (28.9)	51 (14.4)	36 (10.2)	353 (5.1)	
	AFR	283 (72.8)	72 (18.5)	25 (6.4)	9 (2.3)	389 (5.6)	
	EUR	392 (40.2)	378 (38.8)	128 (13.1)	76 (7.8)	974 (13.9)	
Country income	LIC	469 (42.3)	415 (37.4)	135 (12.2)	91 (8.2)	1110 (15.9)	0.004
	LMIC	812 (42.1)	728 (37.8)	278 (14.4)	110 (5.7)	1928 (27.6)	
	UMIC	525 (41.1)	485 (38.0)	146 (11.4)	120 (9.4)	1276 (18.3)	
	HIC	1158 (43.3)	993 (37.1)	341 (12.7)	183 (6.8)	2675 (38.3)	
Sex	Male	1498 (50.2)	1012 (33.9)	309 (10.4)	163 (5.5)	2982 (42.7)	<0.0001
	Female	1466 (36.6)	1609 (40.2)	591 (14.7)	341 (8.5)	4007 (57.3)	
Age	11–14	418 (60.1)	207 (29.7)	55 (7.9)	16 (2.3)	696 (10.0)	<0.0001
	15–17	511 (46.8)	383 (35.0)	149 (13.6)	50 (4.6)	1093 (15.6)	
	18–23	2035 (39.1)	2031 (39.1)	696 (13.4)	438 (8.4)	5200 (74.4)	
Education	Elementary	148 (62.7)	71 (30.1)	15 (6.4)	2 (0.8)	236 (3.4)	<0.0001
	Middle	301 (51.7)	197 (33.8)	68 (11.7)	16 (2.7)	582 (8.3)	
	High school	991 (45.0)	782 (35.5)	289 (13.1)	140 (6.4)	2202 (31.5)	
	University	1524 (38.4)	1571 (39.6)	528 (13.3)	346 (8.7)	3969 (56.8)	
Mother’s education	No education	220 (39.3)	243 (43.4)	65 (11.6)	32 (5.7)	560 (8.0)	<0.0001
	Finished elementary/middle school	583 (39.0)	615 (41.2)	204 (13.7)	92 (6.2)	1494 (21.4)	
	Finished high school	860 (44.0)	702 (35.9)	251 (12.8)	142 (7.3)	1955 (28.0)	
	Finished university education or more	1301 (43.7)	1061 (35.6)	380 (12.8)	238 (8.0)	2980 (42.6)	
Father’s education	No education	127 (40.6)	132 (42.2)	40 (12.8)	14 (4.5)	313 (4.5)	0.001
	Finished elementary/middle school	499 (41.0)	488 (40.1)	163 (13.4)	67 (5.5)	1217 (17.4)	
	Finished high school	721 (41.5)	678 (39.0)	225 (13.0)	113 (6.5)	1737 (24.9)	
	Finished university/more	1617 (43.4)	1323 (35.5)	472 (12.7)	310 (8.3)	3722 (53.3)	
Has medical problems	Yes	233 (25.9)	391 (43.4)	162 (18.0)	114 (12.7)	900 (12.9)	<0.0001
	No	2731 (44.9)	2230 (36.6)	738 (12.1)	390 (6.4)	6089 (87.1)	
COVID-19 infected	Yes	172 (32.5)	215 (40.6)	94 (17.8)	48 (9.1)	529 (7.6)	<0.0001
	No	2792 (43.2)	2406 (37.2)	806 (12.5)	456 (7.1)	6460 (92.4)	
COVID suspected	Yes	388 (30.6)	544 (42.9)	189 (14.9)	147 (11.6)	1268 (18.1)	<0.0001
	No	2576 (45.0)	2077 (36.3)	711 (12.4)	357 (6.2)	5721 (81.9)	
Friends/family COVID-19 infected/suspected	Yes	1060 (34.8)	1263 (41.5)	462 (15.2)	259 (8.5)	3044 (43.6)	<0.0001
	No	1904 (48.3)	1358 (34.4)	438 (11.1)	245 (6.2)	3945 (56.4)	

*p*-value calculated for chi square test.

**Table 2 ijerph-19-10538-t002:** Association between country level factors, sociodemographic profile and medical and COVID-19 status and severity of GAD among 11 to 23-year-old individuals from different countries during the second wave of COVID-19 using multilevel analysis (*n* = 6989).

Factors		AOR (95% CI)		
		Mild vs. Normal	Moderate vs. Normal	Severe vs. Normal
Country region	EMR	1	1	1
	AMR	0.59 (0.42, 0.84)	0.59 (0.38, 0.91)	0.82 (0.53, 1.25)
	SEAR	0.69 (0.53, 0.89)	0.72 (0.52, 0.98)	0.70 (0.49, 1.00)
	WPR	0.65 (0.49, 0.88)	0.87 (0.62, 1.22)	0.92 (0.64, 1.32)
	AFR	0.34 (0.25, 0.45)	0.44 (0.32, 0.63)	0.46 (0.32, 0.67)
	EUR	0.86 (0.70, 1.06)	0.96 (0.74, 1.23)	0.86 (0.66, 1.13)
Country income	LIC	1	1	1
	LMIC	1.47 (1.19, 1.81)	1.49 (1.16, 1.92)	1.25 (0.95, 1.65)
	UMIC	1.23 (0.97, 1.56)	1.08 (0.81, 1.44)	1.25 (0.92, 1.69)
	HIC	1.10 (0.92, 1.31)	1.11 (0.90, 1.38)	1.03 (0.82, 1.30)
Sex	Male	1	1	1
	Female	1.35 (1.20, 1.52)	1.39 (1.21, 1.61)	1.33 (1.14, 1.55)
Age	11–14	0.68 (0.55, 0.83)	0.70 (0.55, 0.90)	0.65 (0.50, 0.86)
	15–17	0.81 (0.69, 0.95)	0.89 (0.73, 1.08)	0.76 (0.61, 0.95)
	18–23	1	1	1
Mother’s education	No education	1	1	1
	Finished elementary/middle school	1.01 (0.78, 1.30)	1.14 (0.83, 1.57)	1.00 (0.70, 1.41)
	Finished high school	0.88 (0.68, 1.15)	1.06 (0.76, 1.47)	0.96 (0.67, 1.37)
	Finished university education or more	0.90 (0.68, 1.18)	1.05 (0.75, 1.48)	0.96 (0.67, 1.38)
Father’s education	No education	1	1	1
	Finished elementary/middle school	0.99 (0.72, 1.36)	0.98 (0.66, 1.46)	1.03 (0.66, 1.60)
	Finished high school	1.08 (0.78, 1.50)	1.01 (0.68, 1.51)	1.14 (0.72, 1.78)
	Finished university/more	1.02 (0.73, 1.41)	1.01 (0.68, 1.51)	1.20 (0.77, 1.88)
Has medical problems	Yes	1.53 (1.27, 1.83)	1.65 (1.34, 2.04)	1.75 (1.40, 2.19)
	No	1	1	1
COVID-19 infected	Yes	0.97 (0.76, 1.24)	1.13 (0.86, 1.49)	0.97 (0.72, 1.32)
	No	1	1	1
COVID suspected	Yes	1.35 (1.14, 1.60)	1.25 (1.02, 1.53)	1.48 (1.20, 1.83)
	No	1	1	1
Friends/family COVID-19 infected/suspected	Yes	1.22 (1.07, 1.38)	1.24 (1.06, 1.44)	1.15 (0.98, 1.36)
	No	1	1	1

AOR: adjusted odds ratio, CI: confidence interval, statistically significant at *p* < 0.05.

## Data Availability

The data used in this study is available upon reasonable request from the corresponding author.

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
