# Peer review of "Anxiety among Adolescents and Young Adults during COVID-19 Pandemic: A Multi-Country Survey"

_ijerph, 2022, doi:10.3390/ijerph191710538_

Round 1

Reviewer 1 Report

This article explores an important topic "Anxiety among adolescents and young adults during COVID- 9 pandemic", but it needs some expansion.

The abstract does not accurately state the objective of the research. 

The introduction lacks a rationale for the research question and how the proposed research could contribute to the generation of new knowledge. It is described in the methodological part of the article, that the research was conducted in 25 countries, but only in two countries it was approved by Research Ethics Committee. It is not clear why this decision was taken.

Discussion section can be expanded with comparison of previous studies on this topic.

The conclusions of the paper are short and could be more detailed. The layout of the paper should also be adjusted (the layout of the tables and list of references is not precise).

Reviewer 2 Report

might choose to briefly discuss problem in literature of getting accurate estimates of anxiety proportions in the population.

Reviewer 3 Report

This report presents data on anxiety among nearly 7000 young adults in 25 countries during the COVID-19 pandemic. Associations were noted between mild, moderate and severe anxiety (based on the GAD-7 instrument) and a number of personal-level and country-level factors. Among the former, the likelihood of severe anxiety (self-assessed GAD-7 score of 15) was independently associated, inter alia, with suspected but unconfirmed COVID-19 infection, however not with COVID-19 status. At the country-level, respondents from the African region were the least likely to be classified as severe anxiety.

I invite the authors to consider the below comments (chronologically ordered).

General Comment:  The paper requires considerable English editing and general editing. By way of example, in the Discussion section (lines 304-305): “This finding was follows previous findings during the pandemic showing that showed there was no association between age and COVID-19 related anxiety”.

ABSTRACT

A survey collected data from 25 countries

A more precise statement would be: A survey collected data from participants in 25 countries. If the word limit is the issue, there are numerous places to trim words. For example: “The data collected were the dependent variables…” could be rephrased as “Dependent variables included…”; “There were 6989 respondents of which 2964…” could be rephrased as “Of the 6989 respondents, 2964…”

“…to determine the association between the dependent, independent and covariate variables

Covariates are also independent variables. Each research question posits different independent variables as the variable/s of interest and all others as “covariates”.

Participants from Eastern-Mediterranean region (EMR) and from lower-middle income countries (LMICs) had significantly higher odds of higher levels of anxiety

Addressing “higher levels of anxiety” is misleading. The sentence is accurate only for moderate vs normal, but not for severe vs. normal. As presented in Table 2, for severe anxiety, the OR for LMIC was not significantly different from the reference category (LIC), and only one region (AFR) was significantly different from EMR (the reference category). A more precise summation of the results is warranted.

Also, AYA in LMICs and EMR had higher risk of having more severe anxiety, emphasizing the importance of public health services in these areas.

This concluding sentence is a repeat of the sentence presented in the previous comment. Besides the inaccuracy (see previous comment), it is not a very helpful conclusion given that over 50 countries are classified by the World Bank as LMIC in which live more than 3 billion people [https://datahelpdesk.worldbank.org/knowledgebase/articles/906519]. 

Also, “more severe anxiety” might be misinterpreted as more clinically severe anxiety, which was not measured (or at least not reported in this paper).

INTRODUCTION

Anxiety disorders refer to excessive apprehension, distress, fear and comparative behavioral disturbances

As there are several types of anxiety disorders, please specify that this refers to generalized anxiety disorder.

What is meant by “comparative behavioral disturbances”?

Please provide the source for this definition.

“…variations in prevalence according to geographic area, cultural differences…”

It is worth mentioning what “cultural differences” refers to.

AYA’s quality of life…

Although this acronym is spelled out in the Abstract, it should be spelled out again here as it is the first time it appears in the body of the report.

“…decreases the utilization of oral health care access

 The word “access” is superfluous.

A meta-analysis showed that 20.5% of children and adolescents experienced anxiety during COVID-19; a figure that doubled that before the pandemic [16]

The 20.5% pooled anxiety prevalence comes from a meta-analysis of 14 studies from China, 4 from the USA and 9 other countries, none of which are Finland. Nonetheless, the authors of that meta-analysis cite a single Finnish study (Tiirikainen et al., 2019) for the pre-COVID prevalence and conclude that “youth mental health difficulties during the COVID-19 pandemic has likely doubled”. I suggest finding more relevant sources for pre-COVID (and possibly even for COVID) anxiety prevalence, preferably from countries included in the present report.

Also, little information is available about whether country level factors affect the risk for anxiety among AYA during the COVID-19 pandemic

It is not readily clear how distal factors such as a country’s GNI or geographic location might be related to anxiety level of young adults. The authors are invited to expound a bit about what country levels factors they are referring to and why they are important to anxiety levels during the pandemic.

MATERIALS AND METHODS

Participants who selected the first two age groups were directed to a parental consent form and after it was filled, the parent was asked to invite the child to respond to the survey without supervision and to assure him/ her of the anonymity and confidentiality of the responses.”

One of the central potential biases inherent in anonymous internet-based surveys, is the true identity of the respondent. Here, not only is the respondent’s true identity unknown, but so too is the parent’s identity. I can readily see a minor completing the parental consent form without the knowledge of his/her parent. The authors are invited to address this concern and state how, if at all, they assured the ethical compliance of minors in this survey.

The second section asked about the participants’ medical history. Participants were required to answer a ‘yes’ or ‘no’ if they had a listed medical problem, had confirmed COVID-19 infection… The COVID-19 questions were adopted from that used for the Mental Health and Wellness Study [20]

Preferably, the entire questionnaire should be made available to the readers. At the very least, the medical conditions should be itemized, especially since this survey seems to have primarily been focused on dental/oral health issues (based on the dental affiliations of most of the authors and the just-published (congratulations!) paper by Tantawi et al on Oral manifestations…

The actual wording of the questions about COVID-19 infection should be stated. In the cited paper, from which the COVID-10 questions were adopted, the questions were: “Respondents were asked if they had tested positive for COVID-19, had COVID-19 symptoms but did not test, had a close friend who tested positive for COVID-19 and/or knew someone who died from COVID-19”.  The data collection period for the present study began in August 2020, which means that a respondent may have been infected/symptomatic a half a year or more prior to the survey. It is not therefore surprising that no association was found between COVID infection and any level of anxiety.

I assume that too few respondents reported knowing someone who died from COVID and this variable was therefore not included in the present report.

The GAD-7 had been validated for use in many of the countries included in this survey with the Cronbach alpha ranging from 0.84 to 0.946

A reference for this should be included. 

“…multi-level multinomial logistic regression model was used to assess the association between the dependent variable (mild, moderate and severe versus normal anxiety) and independent variables (personal, medical and COVID-19-status, WHO region and country income level)

Details about how the models were constructed should be provided.

Significance was set at 5%.

Given that more than 60 comparisons were tested, did the authors consider setting a more conservative significance level to account for multiple comparisons?

RESULTS

The first sentence of the 2nd paragraph (lines 203-208) contains way too much information for a single sentence and should be broken into more manageable chunks.

The word “higher” in “…university education than higher, middle and elementary school education” should be “high school”.

A table listing the proportion of respondents who reported each of the medical conditions would be useful.

Table 1:

Anxiety/GAD is missing from the title.

The percent values in the Total column add up to 100% vertically, whereas in the GAD columns they run horizontally. This is confusing. If the authors are adamant about presenting both, perhaps a blank column should be inserted between the Severe and the Total columns.

The GAD columns requires a heading.

p-values: It is not clear what comparison the p-values presented in the table are referring to. For example, the p-value for country income is 0.004. In the text this p-value is noted for a single comparison: “…from UMICs than from LMICs (9.4% and 5.7%, p=0.004)”. A footnote explaining the p-value should be added.

The asterisk indicating p<0.05 is not necessary since the actual p-value is presented.

Table 2 shows the multi-level multinomial logistic regression model for factors asso-219 ciated with GAD severity levels. Compared to participants from the EMR, participants from the AFR had significantly lower odds of having mild (AOR= 0.34, 95%CI: 0.25, 0.45), moderate (AOR= 0.44, 95%CI: 0.32, 0.63) and severe (AOR= 0.46, 95%CI: 0.32, 0.67) GAD than normal anxiety, while the odds of having mild, moderate and severe GAD than normal anxiety were not significantly different for participants from EURO compared to those from EMR countries”.

Given that the OR and CI values are presented in the table, perhaps they do not need to be repeated here in the text. The readability of the text would be enhanced without all the details.

It seems to me that the finding that the odds of mild anxiety in all regions other than Europe were significantly lower than EMR, would be worth mentioning. 

Having medical problems was significantly associated with mild (AOR= 1.53, 95%CI: 1.27, 1.83), moderate (AOR= 1.65, 95%CI: 1.34, 2.04) and severe (AOR= 1.75, 95%CI: 1.40, 2.19) anxiety compared to normal showing a gradient with a dose response relationship with stronger association in more severe forms of GAD

Was the “dose-response” tested for trend?

DISCUSSION

The study showed that AYA in EMR countries had more severe levels of GAD… higher levels of GAD

Similar to my comment above about the possible misinterpretation of “more severe levels”, perhaps the wording can be changed to “greater probability” or another phrasing that more precisely reflects the intention of the text.

In addition, the study assessed the impact of country level factors on anxiety levels, thus offering a wider perspective for this problem that looks beyond individual factors

Please refer to my above comment about how country-level factors might be related to anxiety.

First, AYA in EMR had higher odds of having severe levels of GAD than participants in other regions.”

Actually, the only comparison that reached statistical significance was with Africa. Also, if the authors wanted to focus on the higher odds among EMR participants, EMR should not have been chosen as the reference category. A more direct summary of the findings in Table 2 would be that AFR had lower odds of severe levels of GAD than EMR.

Also, this study showed that all HICs, UMICs and LMICs had higher levels of mild, moderate and severe general anxiety when compared to normal anxiety than in LICs”

The analyses are based on the averages of countries in the different variable categories. Therefore, the word “all” is imprecise and should be deleted.

The wording “than in LICs” is confusing.

…thus causing a lot more anxiety

‘A lot more” is not appropriate for a scientific report. Please replace with a more quantified assessment.

COVID-19 lock-down may have had more severely affected the economics of LMICs pushing them to poverty and exposing them to severe mental distress Furthermore, COVID-19 pandemic could have affected the care of patients with mental illnesses

The effect of economic downturn within less than one year from the start of the pandemic, especially on young adults in the age bracket captured in this survey, is probably not too strong. Also, do the authors have an estimate of the proportion of 11-23 years olds who have mental illness who do and don’t seek care? I wonder whether these are plausible explanations for the findings presented.

Medical problems make people more vulnerable to serious complications of COVID-19 and this may explain the higher chances of greater anxiety

Few of the underlying conditions associated with a bad COVID prognosis, are prevalent in adolescents. Again, knowing what medical conditions were reported would shed light on this issue.

AYA resident in LMICs and EMRO were at significantly higher risk of having anxiety

As noted above, no significant association was found for country income and only Africa had a significantly lower odds of severe anxiety compared with EMR.

“EMRO” should be corrected to EMR

APPENDIX A

The list is ordered alphabetically by country. Perhaps a more informative presentation would be to group the countries income level.

REFERENCES

Ref 12: All words in the journal name should be capitalized.

Ref 28: The year of publication for this reference is 2022. [citation from PubMed: GBD 2019 Mental Disorders Collaborators. Global, regional, and national burden of 12 mental disorders in 204 countries and territories, 1990-2019: a systematic analysis for the Global Burden of Disease Study 2019. Lancet Psychiatry. 2022;9(2):137-150. doi:10.1016/S2215-0366(21)00395-3]

Author Affiliations

The department and college names for author 19 should be capitalized.

Round 2

Reviewer 3 Report

The authors did a commendable job of responding to the reviewers' comments and questions and revising the manuscript accordingly. 

Minor editorial issues should be corrected. For example:  "Both t ypes were introduced as fixed effect factors. Countries were introduced as random effect factors with intercepts. Robust estimation was used to handle violations to model assumptions. We used least significance difference ti adjust for multiple testing"